# Programmable Electrofluidics for Ionic Liquid Based Neuromorphic Platform

**DOI:** 10.3390/mi10070478

**Published:** 2019-07-17

**Authors:** Walker L. Boldman, Cheng Zhang, Thomas Z. Ward, Dayrl P. Briggs, Bernadeta R. Srijanto, Philip Brisk, Philip D. Rack

**Affiliations:** 1Department of Materials Science and Engineering, University of Tennessee, Knoxville, TN 37996, USA; 2Materials Science and Technology Division, Oak Ridge National Laboratory, Oak Ridge, TN 37831, USA; 3Center for Nanophase Materials Sciences, Oak Ridge National Laboratory, Oak Ridge, TN 37831, USA; 4Department of Computer Science and Engineering, University of California, Riverside, CA 92521, USA

**Keywords:** electrowetting, neuromorphic, ionic liquid, biasing, device, platform, transistors, TFT, TFTs, IGZO, indium gallium zinc oxide, microfluidics, electrochemical, electrostatic

## Abstract

Due to the limit in computing power arising from the Von Neumann bottleneck, computational devices are being developed that mimic neuro-biological processing in the brain by correlating the device characteristics with the synaptic weight of neurons. This platform combines ionic liquid gating and electrowetting for programmable placement/connectivity of the ionic liquid. In this platform, both short-term potentiation (STP) and long-term potentiation (LTP) are realized via electrostatic and electrochemical doping of the amorphous indium gallium zinc oxide (aIGZO), respectively, and pulsed bias measurements are demonstrated for lower power considerations. While compatible with resistive elements, we demonstrate a platform based on transitive amorphous indium gallium zinc oxide (aIGZO) pixel elements. Using a lithium based ionic liquid, we demonstrate both potentiation (decrease in device resistance) and depression (increase in device resistance), and propose a 2D platform array that would enable a much higher pixel count via Active Matrix electrowetting.

## 1. Introduction

Conventional computational devices are designed after Neumann’s model, in which preprogrammed circuits receive and transfer input repeatedly between the memory unit and the logic unit, and subsequently output information based upon the logic unit in the programming. This model has a limited increase in computing power due to the von Neumann bottleneck, which arises because the device cannot both fetch instructions and perform data operation at the same time. To overcome this, neuromorphic computing concepts are being developed that use analog devices to mimic the neuro-biological information processing that occurs in the brain [1,2,3]. Traditional neuromorphic concepts modify the device resistance, forming a correlation between device characteristics and the synaptic weight of neurons. Here, based on our previous work on ionic liquid (IL)-aIGZO electrochemical and electrostatic modulation [4,5,6], we demonstrate a platform via ionic liquid gating of thin film transistors and show that the integration of electrowetting elements can enable the ability to program the position/connectivity of the ionic liquid (IL).

IL gating of metal oxide semiconductor devices has been studied extensively because of the intriguing properties of the electric double layer at the IL/solid interface. This electric double layer generates an extremely high electric field and induces large carrier densities at its interface. This generated electric field has been used to tune the electric properties via carrier concentration modulation in VO_2_ [7], Bi_2_Se3_3_ [8], few-layer graphene [9], and the magnetic properties of certain materials [10,11]. The electric field can also extract or intercalate ions out of or into the solid, respectively. Intercalation of hydrogen and hydroxyl ions via a hydrated IL has been observed in WO_3_ [12], ZnO [13], and MoO_x_ [14], and intercalation of O_2_ has been observed in SrTiO_3_ [15], WO_3_ [16], and TiO_2_ [17], among others. Electrostatic electron doping [4] and electrochemical oxygen extraction [5] has also been demonstrated in amorphous indium gallium zinc oxide (aIGZO) thin film transistors (TFTs). Combining these phenomena in an aIGZO transistor active layer allows us to tune the electrical and chemical properties, creating both short-term potentiation or plasticity and long-term potentiation or plasticity. Short-term potentiation (STP) is generally defined as a brief, reversible, and repeatable change in synaptic strength between two neurons, usually in the range of a few milliseconds to a few seconds. Here, STP is demonstrated via the electrostatic effects of the IL. Long-term potentiation has a greater, more permanent effect on synaptic strength, and is realized here via the electrochemical doping of the aIGZO TFTs. While neuromorphic computing platforms rely on, for instance, crossbars of variable resistive states, we utilize, for convenience, changes in the aIGZO transfer characteristics to demonstrate the programmable platform. Note that resistive elements are easily integrable with the proposed platform.

## 2. Materials and Methods

In this study, we combined the electrical tuning of a TFT active layer with a pixelated electrowetting on dielectric (EWOD) array as a programmable neuromorphic device platform. Figure 1 overviews (a) the device architecture and (b–f) illustrates the various functionalities of the platform. As shown in Figure 1a, the device architecture consisted of a top and bottom plate. The bottom plate contained a 2D array of pixels, where each pixel contained an integrated dual gate thin film transistor and an electrowetting on dielectric electrode. The top plate was a pixelated electrode array, which acted as the top gate for the long-term potentiation of the device. See Appendix A for a detailed description of the device synthesis. A key feature of the device is the programmability of the IL network over the 2D pixel array via electrowetting. As illustrated in Figure 1b–e, electrowetting of the IL was accomplished via activating the (b) left pixel, which contained the IL, and thus inducing a lowering of the hydrophobicity of the IL, which caused spreading. (c) By activating the right pixel, spreading was directed to this pixel. After the IL had spread to this right pixel, the left pixel was turned off (d), thus increasing the hydrophobicity, which caused the drop to dewet from the left pixel. Clearly, any neuromorphic application will require long-term electrowetting stability. Interestingly, while many electrowetting lab-on-chip applications have been proposed, relatively little work has been devoted to studying long-term stability of these devices. In a study done by Dhindsa et al., a study was done with a dip-coated fluoropel layer where over 300 cycles were achieved without any significant change to the contact angle or degradation of the hydrophobic layer [18]. Similarly, work done by Mibus and Zangari showed that, for a TaOx/spin coated Cytop insulator/hydrophobic layer stack, stable electrowetting of up to 350 cycles was achieved at a 20 V bias [19]; however, severe degradation occurred at 25 V applied bias. As will be demonstrated, the presence of the IL on the aIGZO active layer changed the transfer characteristics and thus realized STP. LTP was realized by applying a top gate bias (f) to the IL, which electrochemically extracted/intercalated ions out/into the active layer and changed the electrical properties of the aIGZO. See Appendix A for the full device concept video.

## 3. Results

Figure 2a shows back gate transfer characteristics with and without the presence of IL on the active layer, which illustrates that STP (volatile memory) was realized via electrowetting the IL over the desired aIGZO TFT pixel. Note: Current values below 1e-14 amps were truncated due to instrument sensitivity. For this experiment, diethylmethyl (2-methoxyethyl) ammonium bis(trifluoromethyl sulfonyl)imide (DEME) was chosen as the ionic liquid. The TFT transfer characteristics changed dramatically, as noted by an ~580× increase in the *I*_ds_ when *V*_g_ was equal to zero. This electrostatic effect was only present while the IL/solid interface was formed, and could thus be combined with electrowetting of the IL to create a programmable STP device. Combining this effect with an active matrix array of transistors [18,19] could result in a neuromorphic platform that could be scaled to very high pixel counts.

Non-volatile memory, or LTP, was induced by electrochemically altering the chemical composition in the active layer via a positive top gate bias. To characterize this LTP effect, Figure 2b–d illustrates a series of electrical measurements taken on the IGZO active layer. For this experiment, both DEME and LiClO–PEO were used. DEME electrochemically dopes the IGZO active layer by extracting oxygen ions under a positive gate bias, and thus decreasing the device resistance. The electrochemical effect of the DEME is permanent but, unlike several other oxide materials, is non-reversible. To create a device capable of both potentiation (decrease in device resistance) and depression (increase in device resistance), the electrostatic and electrochemical effects of DEME were compared to another ionic liquid, LiClO–PEO. Rather than extracting oxygen, LiClO–PEO intercalates (positive gate bias) or extracts (negative gate bias) lithium ions into the near surface region of the aIGZO active layer, which in turn decreases or increases device resistance, respectively [20,21,22]. To demonstrate the LTP, after the ionic liquid is actuated over the pixel, a top gate bias of +2 V was applied to the IL/active layer for various time intervals (10, 10, 30, 50, and 100 s) for a total of 200 s (see *I*_ds_ measurements during top gating in Appendix A). Figure 2b,c show the back gate transfer characteristics for the DEME and LiClO–PEO ILs (respectively) measured in between each time interval, where the ILs induce higher conductance in the aIGZO channels. After a cumulative bias time of 200 s, a negative bias (*V*_g_ = −2 V) was applied to both devices, indicated by the dashed line in Figure 2b,c; note that the DEME channel inexplicably continued to decrease the channel resistance, whereas the −2 V bias for the LiClO-PEO IL increased the channel resistance. To demonstrate the purely electrochemical change, Figure 2d illustrates the currents at *V*_g_ = 0 V for each transfer measurement plotted as a function of top gate bias time and normalized with respect to the current at zero gate voltage before biasing (note that off state current for the DEME is much lower than that of the LiClO–PEO). For both ILs, the electrochemical effect of positive biasing tended to saturate after 100 s. TFT resistance under DEME biasing changed by over two orders of magnitude in under a minute, promoting fast writing/programming time in a neuromorphic device. While the electrochemical effect of the DEME was much more pronounced, applying a negative bias further decreases device resistance; the LiClO–PEO IL, however, when biased at −2 V, increased the resistance and thus demonstrated reversible transport modulation.

Since low power computation is a significant driving force for neuromorphic computing, we explored short pulse biasing with the DEME IL. Figure 3a illustrates three series +2 V 50 ms gate bias pulses (15, 50, and 100 pulses, respectively) where the *I*_ds_ (*V*_ds_ = 1 V) was measured during the IL gate bias. Figure 3b is a magnified view of seven pulses of the 100 pulse series. The electrostatic IL effect was clear as the on-current increased over six orders of magnitude. To measure a residual electrochemical effect, after each series of pulses, *I*_ds_ was allowed to decay to a pseudo-saturation point, allowing any remnant electrostatic effect to decay. Figure 3c is a plot of *I*_ds_ after each pulse series, which illustrates that the change in remnant current increased with an increase in the number of pulses. Figure 3d shows the remnant *I*_ds_ current measured two seconds after each pulse, demonstrating a permanent and incremental decrease in the channel resistance.

The proposed programmability of the neuromorphic platform was realized via electrowetting of the IL. Electrowetting has been used in the development of Lab on a Chip [18,23,24], flexible and transparent displays [25], as an aid in medical and chemical analysis [26,27], and novel device concepts, such as an aerosol sampler [28] and a micro conveyor system [29]. The electrowetting of the IL is governed by the Young-Lippmann equation, γ_LG_cos(θ) = γ_SG_ – γ_SL_ + (1/2)CV^2^, where θ is the wetting angle, γ_SG_ is the surface tension between the substrate and air (or oil), γ_LG_ between the IL and air (or oil), γ_SL_ between the solid and IL, C is the capacitance per unit area, and V is the applied electric voltage. As the capacitance of the electrowetting dielectric is critical to the performance, we used a 100 nm SiO_2_ electrowetting dielectric with a specific capacitance of 39.4 pF/cm^2^. The electrowetting behavior of many different ionic liquids have been observed, with voltages required to achieve electrowetting ranging from 25 to 190 V [30,31]. The contact angle saturation in ionic liquids have been studied [30], and the voltage range for saturation is shown to be 40 to 70 V. In an optimized neuromorphic platform, low-voltage electrowetting would be critical for low-power consumption, and thus contact angle saturation should not limit device functionality. Furthermore, to lower the EW voltage, nanostructured EW electrodes could also be employed to enhance the hydrophobicity. Thus, we characterized the basic electrowetting characteristics of DEME via a series of actuation tests of the IL (relative to a standard yellow ink) as a function of increasing voltage, as detailed in the Appendix A. To combine the electrowetting platform with the TFT pixel array, a transparent top electrode plate was patterned via indium tin oxide (ITO) sputtering and lift-off. This top plate acts as a common ground during EW actuation of IL droplet, and as the programmable IL gate electrode during biasing. To form a channel, a DEME reservoir was dispensed onto the electrowetting platform via a micropipette next to the electrowetting array, as shown in Figure 4a. Subsequently, the electrowetting pixels 1–8 were progressively activated by applying a 20 V bias to each pixel. Figure 4b is a still image of the early stage channel formation, where just EWP1 was biased. Figure 4c shows the channel formation after the first five pixels had been activated, which took approximately 10 s for the viscous IL to cover. Full channel formation, Figure 4d, took approximately 24 s (see Appendix A for full video); however, channel retention was only maintained for as long as the bias was applied. Similarly, agile droplet motion is also achievable via sequential pixel biasing, as schematically illustrated in Figure 1 and demonstrated in Figure 4f–h. Initial actuation/biasing of the original IL pixel (Figure 4f) caused the drop to spread and directed spreading was accomplished by simultaneously addressing the appropriate neighboring pixels (g). After the droplet spread over both pixels, de-activating the original pixel (h) caused the drop to dewet from the original pixel (see full video in Appendix A). Full 2D channel formation and droplet motion was possible, leading to a deterministic pattern of the IL on the pixelated array. 

## 4. Discussion

By combining the electrochemical doping of the aIGZO TFTs and the device programmability described above, we could now realize multi-pixel programming, which would lead to an exponential decrease in both programming time and required power. Since the electrochemical gating was applied via a uniform top gate across the entire device (see Appendix A), any and all TFTs with IL present would be biased, thus dramatically reducing the number of bias runs needed to program a full device. Figure 5 shows a conceptual 8 × 8 TFT array demonstrating multi-pixel programming via multiple IL droplets over multiple pixel arrays, with the pixels being programmed highlighted in red. By combining these device concepts with active matrix electrowetting, a programmable electrofluidic neuromorphic platform with very high pixel count could be realized. Multi-dimensional channel formation is envisioned via a large 2D array of pixels, and channel retention without bias (non-volatility) is possible if combined with so-called “Laplace Barrier” posts on the top plate [32].

In conclusion, by mimicking the neuro-biological processing via an ionic liquid neuromorphic platform, we demonstrated both short- and long-term potentiation, as well as reversibility of the device resistance. Short-term potentiation was achieved by electrowetting the IL over the desired TFT/TFTs, and long-term potentiation was achieved by electrochemically doping the TFT. We showed successful doping of the aIGZO via oxygen extraction using DEME, and reversible intercalation of lithium ions using LiClO–PEO. Short pulsed biasing was performed as a way to minimize power. IL channel formation and droplet mobility were both demonstrated and characterized in order to move towards a multi-pixel biasing scheme.

## Figures and Tables

**Figure 1 micromachines-10-00478-f001:**
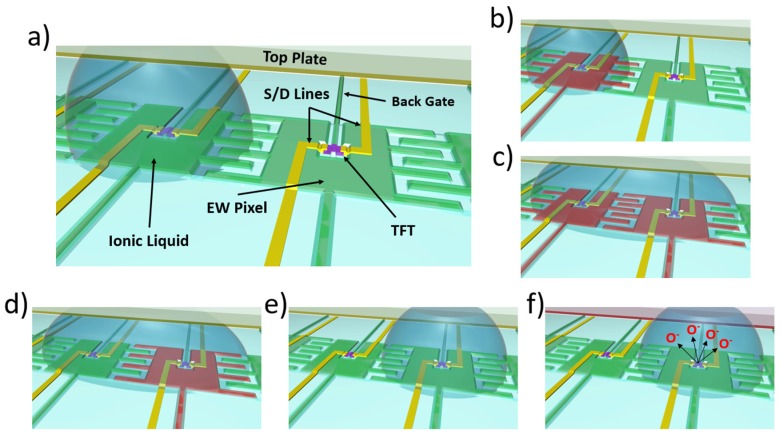
(**a**) Schematic of programmable electrofluidic neuromorphic device, with critical components labeled. See Appendix A for a detailed and labeled 2D cross-section of device. (**b**–**e**) Schematic illustration demonstrating the electrofluidic control of the ionic liquid (IL). In (**b**), the ionic liquid is present initially over the left pixel and is electrofluidically transferred to the neighboring right pixel by initially actuating the left pixel, which decreases the surface energy and causes the IL to spread. Next, (**c**) the neighboring right pixel is actuated, which causes the IL to selectively spread to this pixel. The left pixel is then turned off (**d**), which increases the surface energy, and thus the ionic liquid migrates over to the right pixel. After movement to this pixel, the right pixel is then turned off (**e**), which completes the cycle. Short-term plasticity is realized when a transistor is measured with IL present on the pixel. Long-term plasticity is realized by applying a bias on the ionic liquid via the top plate (**f**), which electrochemically alters the composition of the material by extracting oxygen ions, permanently altering its electrical resistance (see Appendix A for full video).

**Figure 2 micromachines-10-00478-f002:**
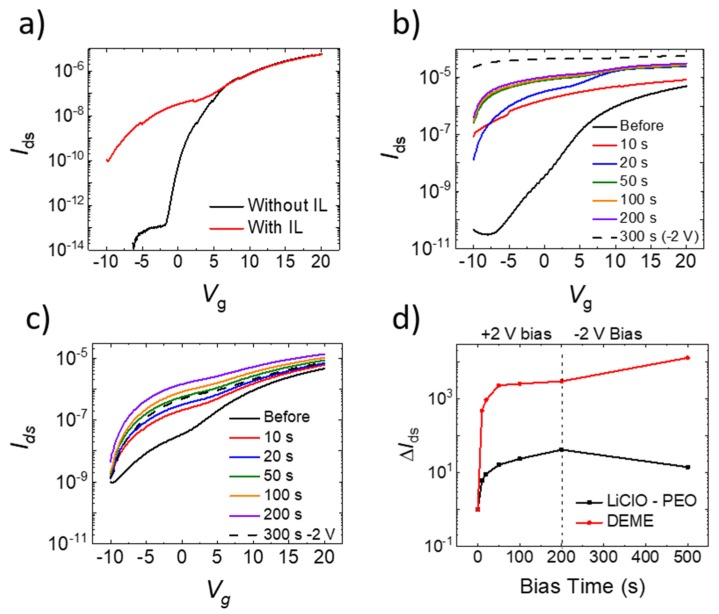
(**a**) Back gate transfer characteristics of a thin film transistor (TFT) pixel both with and without the presence of the DEME ionic liquid. After electrowetting the ionic liquid over the TFT pixel, the transfer characteristics shifts due to the presence of the IL. (**b**,**c**) illustrates the back gate transfer characteristics after each IL bias time, which demonstrates the electrochemical long-term potentiation (LTP) due to biasing of DEME and LiClO–PEO, respectively. After a cumulative 200 s positive bias, a −2 V bias for 300 s was then applied to both ILs in an attempt to reverse the characteristics; note only the LiClO–PEO IL exhibits some recovery of the high-resistance state. The *I*_ds_ at zero gate voltage as a function of IL biasing is shown in figure (**d**) as a function of cumulative bias time. In (**a**–**d**), *V*_ds_ was 1 V.

**Figure 3 micromachines-10-00478-f003:**
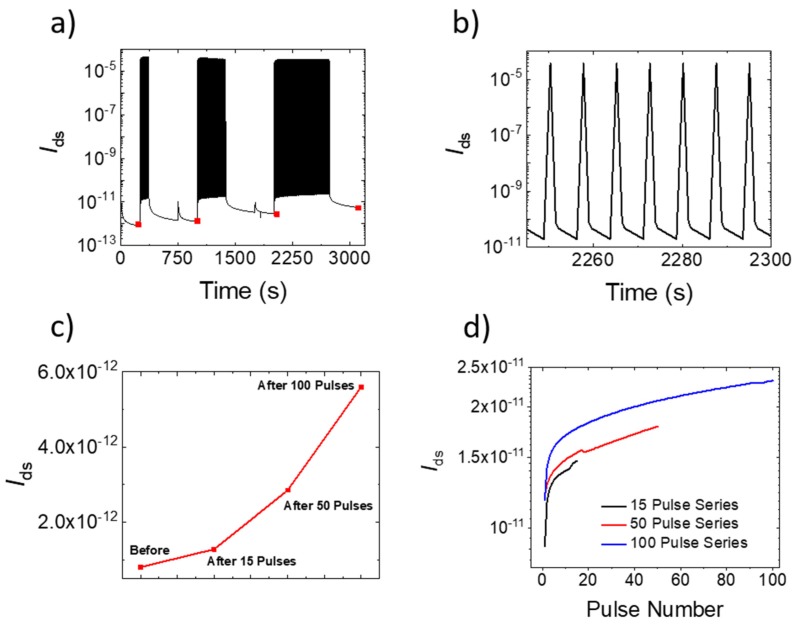
(**a**) *I*_ds_ as a function of time for a series of +2 V, 50 ms top gate pulses (15, 50, and 100, respectively) applied to the active layer. (**b**) Magnified view of seven pulsed cycles from (**a**). (**c**) Remnant *I*_ds_ values (red squares in (**a**)) after each pulse series, where the relaxation time after the last pulse was 630 s. To illustrate the cumulative effect of each +2 V 50 ms bias pulse, the *I*_ds_ versus pulse number is plotted for each series after a relaxation time of 2 s per pulse. In (**a**–**d**), *V*_ds_ was 1 V.

**Figure 4 micromachines-10-00478-f004:**
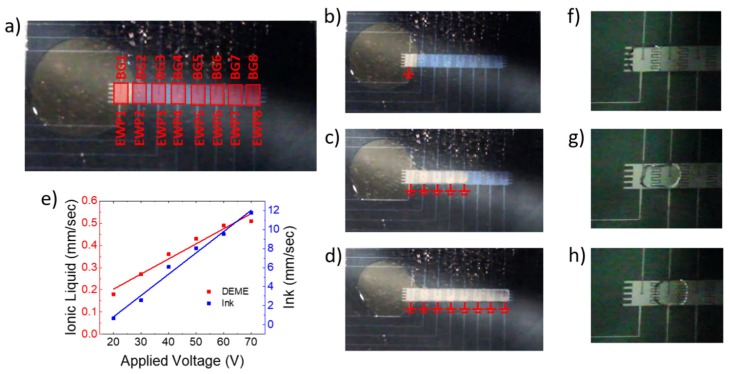
(**a**) Photograph of programmable electrofluidic neuromorphic device with the circular ionic liquid reservoir shown to the left (slight yellow tint) of the 1D electrowetting array. Overlaid in (**a**) are the rectangular pixels and the labels for the back gate contact lines (BG 1–8) and electrowetting pixel contact lines (EWP 1–8). (**b**) Channel formation with only EWP1 turned on. (**c**) Channel formation after EWP1–5 have been powered (10 s) and (**d**) channel formation of IL after all EWP1–EWP8 are powered (24 s). Electrowetting voltage in (**b**–**d**) and (**f**–**h**) was 20 V. (**e**) Plot of electrowetting speeds of DEME compared to a standard aqueous based yellow ink as a function of applied voltage. (**f**–**h**) illustrate IL droplet motion, similar to schematically illustrated in Figure 1 (see text for details).

**Figure 5 micromachines-10-00478-f005:**
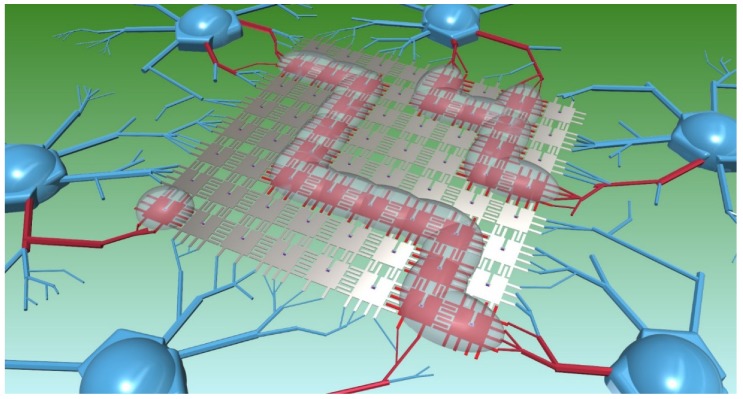
Conceptual model of a 2D array programmable electrofluidic neuromorphic device capable of multi-pixel biasing. Here, three ionic liquid droplets are shaped via electrowetting into user defined channels, allowing specific chosen pixels (highlighted in red) to be programmed.

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
