# Peer review of "Programmable Electrofluidics for Ionic Liquid Based Neuromorphic Platform"

_micromachines, 2019, doi:10.3390/mi10070478_

Round 1

Reviewer 1 Report

The authors present a microfluidics platform that enables the position/connectivity of an ionic liquid. The liquid motion is performed by using electrowetting (EW) phenomenon. The above apparatus is a demonstration (maybe an early one) of an neuromorphic computer.

Before accepting the manuscript, I would like the authors to clarify some issues:

1) Can they give more details about the dielectric thickness as well as capacitance per unit area, C, value?

2) What, according to the authors' opinion, is the life duration of this setup - how many EW cycles can approximately perform till degradation?

3) I would like also a comment regarding the contact angle saturation phenomenon in EW. Will it affect the functionality of the device?

4) Is it possible to use structured (superhydrophobic) substrates in order to reduce the contact angle hysteresis during droplet motion?

Reviewer 2 Report

"The authors presented their development of ionic liquid gating and electrowetting for programmable placement/connectivity of the of the ionic liquid.  My specific comments are as follows. Some more explanation for experimental data might be needed, such as the black-line data shown at Fig 2(a). What happens for the negative Vg?
